# communications
# engineering

# The silent impact of underground climate change on civil infrastructure

Alessandro F. Rotta Loria [1✉]

Urban areas increasingly suffer from subsurface heat islands: an underground climate change responsible for environmental, public health, and transportation issues. Soils, rocks, and construction materials deform under the influence of temperature variations and excessive deformations can affect the performance of civil infrastructure. Here I explore if ground deformations caused by subsurface heat islands might affect civil infrastructure. The Chicago Loop district is used as a case study. A 3-D computer model informed by data collected via a network of temperature sensors is used to characterize the ground temperature variations, deformations, and displacements caused by underground climate change. These deformations and displacements are significant and, on a case-by-case basis, may be incompatible with the operational requirements of civil structures. Therefore, the impact of underground climate change on civil infrastructure should be considered in future urban planning strategies to avoid possible structural damage and malfunction. Overall, this work suggests that underground climate change can represent a silent hazard for civil infrastructure in the Chicago Loop and other urban areas worldwide, but also an opportunity to reutilize or minimize waste heat in the ground.

[1] Mechanics and Energy Laboratory, Department of Civil and Environmental Engineering, Northwestern University, 2145 Sheridan Road, Evanston, IL 60208, USA. ✉email: af-rottaloria@northwestern.edu

The ground beneath urban areas is warming up, leading to subsurface urban heat islands[1]. This underground climate change has two types of causes: anthropogenic and meteorological. The leading cause, developing over timescales of years, consists of thermal perturbations of the underground due to anthropogenic activity. Buildings and infrastructures continuously inject heat into the ground due to thermal losses associated with indoor heating and operating appliances[2–10]. Underground transport repeatedly impacts the temperature field of the subsurface with heat emitted by trains braking, or cars and people traveling[11–14]. Underground pipelines, sewers, high-voltage cables, and district heating systems also heat the ground[15]. Another cause of subsurface heat islands, developing over timescales of centuries, consists of meteorological influences. Rises in air temperature above the ground due to the daily absorption from construction materials of solar radiation and other heat sources are leading to meteorological urban heat islands[16,17]. As the ground temperature (e.g., beyond the shallowest 4–6 m down to 50–100 m) is typically close to the mean annual surface air temperature, and the air temperature is increasing due to urban heat islands, the ground is also warming up. Therefore, subsurface urban heat islands can partly be considered as the underground thermal imprint of meteorological urban heat islands[18].

Proportional to urban density and population[6,19], and dependent on topography and hydrogeology of the urban space[20], subsurface urban heat islands are an alarming phenomenon for urban areas, which can often be more intense than their surface counterpart[21]. A recent review of the literature suggests that subsurface heat islands are causing in various cities across the world an increase in average ground temperature between 0.1 and 2.5 °C per decade down to 100 m of depth[22]. Studies highlight multiple impacts of subsurface temperature rises on urban areas. Subsurface temperature rises can affect the biochemical state[8,23–26] and hydrogeological state[3,6,9,10,21,27–29] of the urban underground, leading to shifts in plant growth and thermal pollution of groundwaters, among other issues. Subsurface temperature rises can also cause transportation infrastructure and public health issues, such as overheated subway rails that force trains to slow down or stop to avoid incidents with significant economic costs associated with the delay of public transportation services, and extreme air temperatures underground that cause thermal discomfort and heat-induced diseases, such as heat cramp, dehydration, hypertension, asthma, and heatstroke[13,30–37]. On the contrary, subsurface temperature rises represent an opportunity, as geothermal technologies can harness and reutilize additional heat from the ground[38–44].

The fundamental hypothesis behind this work is that subsurface heat islands represent a silent hazard for urban areas, with detrimental possibilities for the performance of civil infrastructure. This hypothesis relies on three considerations: (1) soils, rocks, and construction materials are affected by temperature variations, undergoing thermally induced deformations and property changes that can be reversible or irreversible over time[45,46]; (2) the average temperature of the shallow subsurface in urban areas is rising at an alarming rate, with recorded ground temperature anomalies in the core of dense city districts that can achieve up to +20 °C[6,9,15,22]; (3) comparable temperature variations to those that are currently measured in the subsurface of urban areas have shown to represent an issue for the geotechnical and structural performance of geothermal structures and infrastructures, and for this reason, must now be considered in their design[47,48]; however, no existing civil structure or infrastructure in cities has been designed to account for rising ground temperatures and is hence prone to operational issues due to subsurface heat islands.

Motivated by the lack of a fundamental understanding of the impacts of subsurface heat islands on the performance of civil infrastructure, this study addresses such knowledge gap and validates its underlying hypothesis with reference to a real case study: the Chicago Loop district—the most densely populated district in the US after Manhattan, which suffers from an urban heat island[22]. Two facilities are used to explore this complex problem: a 3-D computer model of the Chicago Loop and a wireless temperature sensing network installed in surface and subsurface environments across such district.

The developed computer model reproduces the urban morphology of the Loop with due account of the building basements, underground parking garages, subway tunnels, train stations, and freight tunnels that characterize such a district. Based on a substantial amount of temperature data gathered from underground built environments and the ground surface, the model allows for the simulation of the waste heat continuously injected into the ground (see "Methods, Temperature sensing network"). The employed simulation approach consists of 3-D, time-dependent, thermo-hydro-mechanical finite element modeling, which not only allows to quantify the temperature variations that characterize the subsurface of the Loop in space and time but also their effects on its deformation and the groundwater flow (see "Methods, Numerical model and simulation").

Simulations are performed over 100 years: from 1951, when the subway tunnels in the Loop were completed and the morphology of its underground built environments approached the current state, till 2051. The simulation results provide ground temperature values that match with recent data collected from the heart of the Loop's subsurface (see "Methods, Numerical model validation"). On the one hand, this evidence allows retrieving the evolution of the temperature field across the Loop from the 1950s to date. On the other hand, this result allows for the prediction of temperature rises that are likely to develop over the next thirty years in the subsurface of the Loop. Jointly, the results provide a quantification of the thermally induced ground deformations and displacements resulting from subsurface urban heat islands considering the Loop.

Based on the results of this study, the impacts of temperature variations associated with subsurface heat islands are shown to represent a silent hazard for the operational performance of civil infrastructure in Chicago and potentially other cities worldwide. Considering this issue, the need to revise current urban planning strategies to mitigate subsurface urban heat islands is finally discussed for Chicago and other cities considering two strategies detailed in this work.

## Results

**Temperature variations caused by the underground climate change in the Loop.** Figure 1 shows the evolution of the ground temperature in the Loop from the 1950s till the 2050s. Specifically, Fig. 1a illustrates the temperature field that characterizes the Loop at depths of $z = 10$, 17.5, and 23 m (i.e., corresponding to the average depths of the soft, stiff, and hard clay layers underneath such district, respectively) and after simulation times of $t = 1$, 71, and 100 years (i.e., corresponding to the years 1951, 2022, and 2051, respectively); Fig. 1b illustrates the average temperature trends for all the soil layers underneath the considered district (the average ground temperatures are calculated over the volume constituting the relevant soil layer, whereas the weighted ground temperature is an average of these values; weighting is applied with respect to the thickness of the soil layers).

As can be noted in Fig. 1a, the temperature field in the subsurface of the Loop is highly non-uniform because of the heat

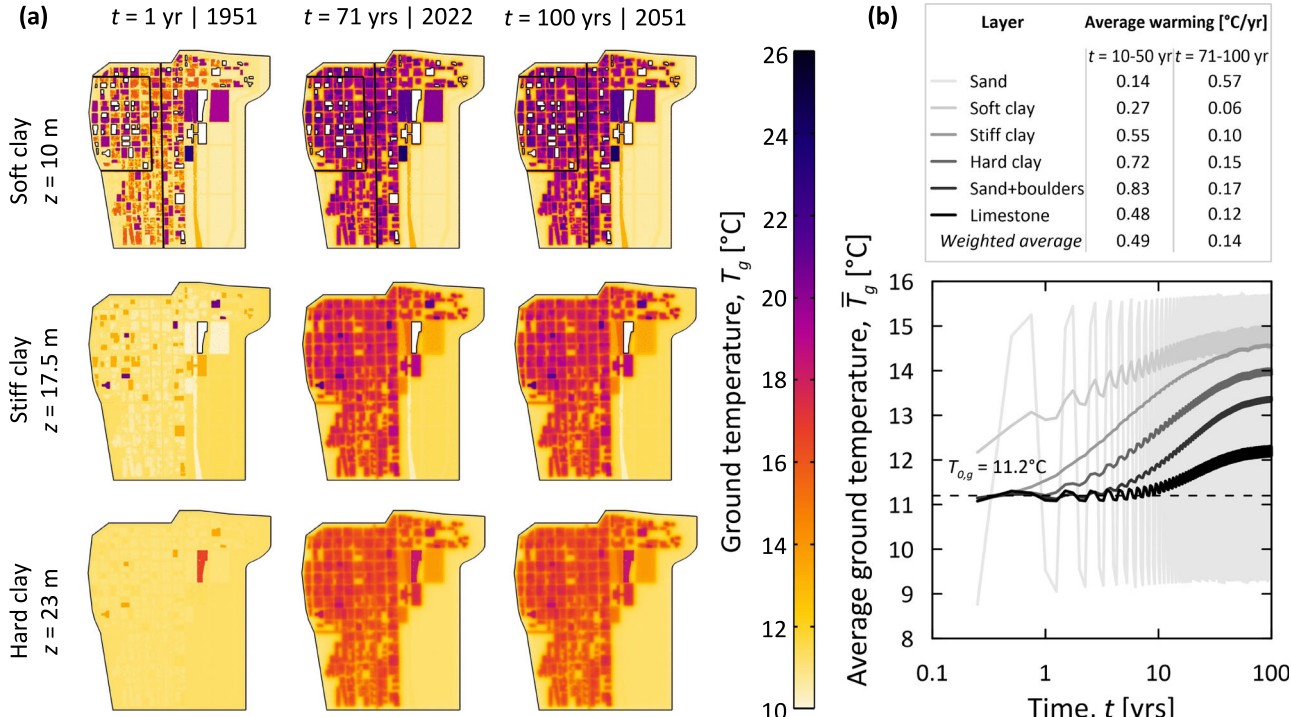

**Fig. 1 Temperature anomalies underneath the Chicago Loop. a** Ground temperature distributions, $T_g$, at varying depths, $z$, and times $t$ (the darker the shading, the higher the ground temperature); **b** average temperature trends, $\bar{T}_g$, for individual soil layers from the initial value of $T_{0,g} = 11.2\,°C$ (displayed with the black dashed line).

rejected from underground built environments and the ground surface. The magnitude of ground temperatures markedly depends on the morphology of the underground built environments that characterize the Loop district, consistent with previous evidence[6,9,10,14,15,20,42]. Ground temperature variations are more significant where underground built environments get denser and decrease with increasing distance from such environments (see "Methods, Temperature sensing network" for details about the morphology of the Loop). Accordingly, larger ground temperature variations characterize the northern (i.e., more densely built) portion of the Loop compared to the southern (i.e., less densely built) portion of such district. The ground in the south-eastern portion of the Loop across Grant Park presents negligible temperature variations at depth due to the absence of waste heat sources. The ground in the north-eastern portion of the Loop across Millennium Park presents temperature anomalies due to the presence of Millennium Garages, the underground train station operated by METRA, and the Harris Theater for Music and Dance. Further north from this area, the Loop presents relatively limited ground temperature variations due to a coarse distribution of buildings and the presence of Lakeshore East Park. Ground temperature variations can be very significant when considered locally, with values that can exceed $\Delta T_g = 15\,°C$. Meanwhile, ground temperature variations are limited when averaged in space within distinct soil layers, with values of about $\Delta \bar{T}_g = 1–5\,°C$.

As can be noted in Fig. 1b, the average ground temperature underneath the Loop has significantly increased over the past 70 years, whereas it currently appears to be in a thermal quasi-steady state (i.e., slowly approaching a thermal saturation). In the past, an annual average ground warming of 0.49 °C/yr has characterized the Loop down to 100 m of depth. Currently, the ground is warming at an annual rate of 0.14 °C/yr. Different soil layers undergo a variable temperature variation over time depending on the presence, geometry, and density of heat sources, the distance

from the ground surface, the ground thermo-physical properties, and the presence and magnitude of groundwater flow. Temperature variations affect soil layers starting from different time-frames, with shallower layers warming up faster than deeper layers because of the time required for the heat rejected by underground structures to reach greater depths. Ground temperature variations are mostly affected by the surface thermal conditions at shallower depths (showing the greatest sensitivity in correspondence of the top sand layer) and become decreasingly affected by such conditions as the depth increases from the soft clay towards the limestone layers (see "Methods, Numerical model and simulations" for details about the soil stratigraphy). The significant thermal conductivity of the sand and boulders layer and the limestone bedrock involves remarkable warming of these layers when they start to be affected by the heat rejected by underground built environments. Currently, the limestone bedrock appears to undergo a comparable, if not more significant, warming than shallower layers (with the exception of the top sand layer, which is markedly influenced by the surface thermal conditions), despite its greater distance from the sources of waste heat. This evidence derives from the fact that the thermal state of the limestone bedrock is further away from a thermal saturation condition compared to the shallower soil layers.

**Deformations caused by the underground climate change in the Loop**. Figure 2 shows the impacts of the underground climate change of the Loop on its deformation state from the 1950s till the 2050s. Specifically, Fig. 2a illustrates the thermally induced deformation that characterizes the Loop in the middle of the soft, stiff, and hard clay layers after $t = 1$, 71, and 100 years; Fig. 2b illustrates the average vertical displacement trends for all the soil layers underneath the Loop (the average ground displacements are calculated over the volume of the relevant soil layer), together with the maximum vertical displacement values for such layers.

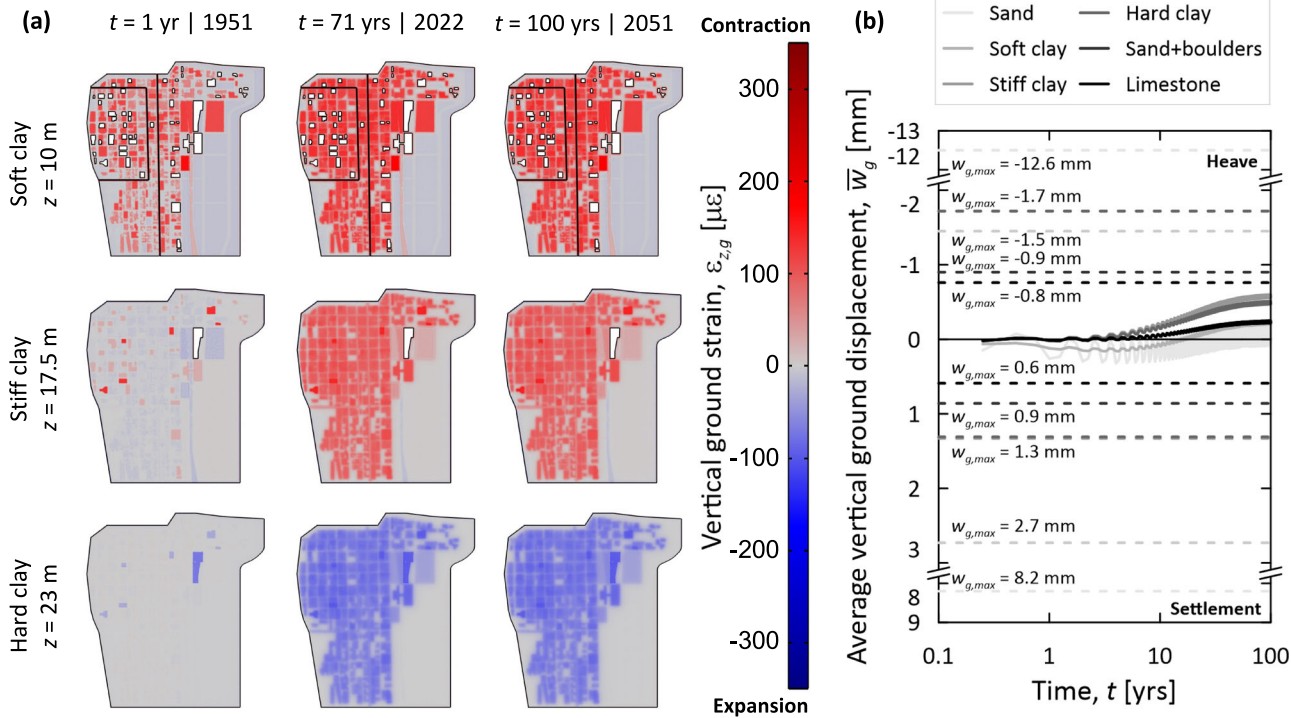

**Fig. 2 Influence of temperature anomalies on the deformation of the Chicago Loop. a** Vertical strain variations in the ground, $\varepsilon_{z,g}$, at varying depths, $z$, and times $t$ (the red shading indicates contractive strains; the blue shading indicates expansive strains); **b** average vertical displacement trends, $\overline{w}_g$, for individual soil layers ($w_{g,max}$: maximum vertical displacement values displayed with the dashed lines).

As can be noted in Fig. 2a, consistent with the features of the temperature field and the coupling between temperature and strain, the vertical deformation field is markedly non-uniform. The observed vertical strains are contractive for the soft and stiff clay layers, whereas they are expansive for the hard clay layer due to their respective consolidation states (normally consolidated vs. overconsolidated)[45]. The shallower and deeper sand layers and the bottom limestone layer expand due to the observed temperature rises. The greatest strains are observed in correspondence with the largest temperature anomalies. Depending on the considered layer, thermally induced vertical strains of up to $\varepsilon_{z,g} = \pm 300\ \mu\varepsilon$ (i.e., $\varepsilon_{z,g} = \pm 300 \times 10^{-6}$) characterize the ground.

As can be noted in Fig. 2b, the thermally induced vertical strains of the ground are accompanied by vertical ground displacements. The average values of such displacements may be considered limited from a practical perspective, as they generally remain in the range $w_g = \pm 1$ mm. However, vertical displacements can be significant in localized ground zones close to sources of waste heat. Maximum thermally induced heaves (i.e., upward displacements) can exceed $w_{g,max} = -12$ mm. Maximum thermally induced settlements (i.e., downward displacements) can exceed $w_{g,max} = +8$ mm. Vertical displacements of several millimeters characterize the soil layers embedding the largest number of heat sources. Both settlements and heaves can characterize an individual soil layer because of the complex deformation patterns that are renowned to characterize the response of underground structures and excavations subjected to temperature variations[48]. Differential displacements of several millimeters are found underneath various buildings across the Loop, close to the boundaries (e.g., walls and slabs) where heat is injected into the ground.

**Underground climate change: a silent hazard that can represent a resource**. Vertical ground displacements of the order of millimeters can affect the operational performance of foundations and earth retaining structures, as they can fully mobilize the shaft capacity of piles and induce excessive rotations, tilt, and deflections of structural members, such as walls and slabs[48–50]. Differential displacements, rather than average displacements, generally are the primary cause of concern for the performance of foundations and the actual damage to superstructures, which can affect both their visual appearance and function[51,52]. In the context of subsurface urban heat islands, differential and average displacements not only depend on the geometrical and structural features of the foundation and the superstructure, the properties of the building materials, the construction details and finishes, and the properties of the ground but also on the spatial distribution of the temperature variations caused by waste heat emissions.

As the underground climate change that has affected the Loop in the past has led to vertical displacements of several millimeters due to waste heat rejected by underground built environments, such phenomenon might have silently contributed to some of the documented operational issues for buildings and infrastructures in such district[53–55]. These include excessive settlements of foundations, visible deflections of structural elements from the vertical or horizontal, and cracking of structural members with interconnected durability issues for several buildings constructed in the Loop after the Chicago fire in 1871 and through the 1900s. Historically, these issues have been attributed to inappropriate foundation designs and construction methods. However, groundward heat diffusion driven by the operation of basements and underground facilities might have exacerbated these issues.

As the ground underneath the Loop is currently undergoing limited temperature rises, small thermally induced ground displacements are likely to develop in the years to come. However, the ongoing underground climate change should be mitigated to avoid unwanted impacts on civil structures and infrastructures in the future, as currently acceptable ground

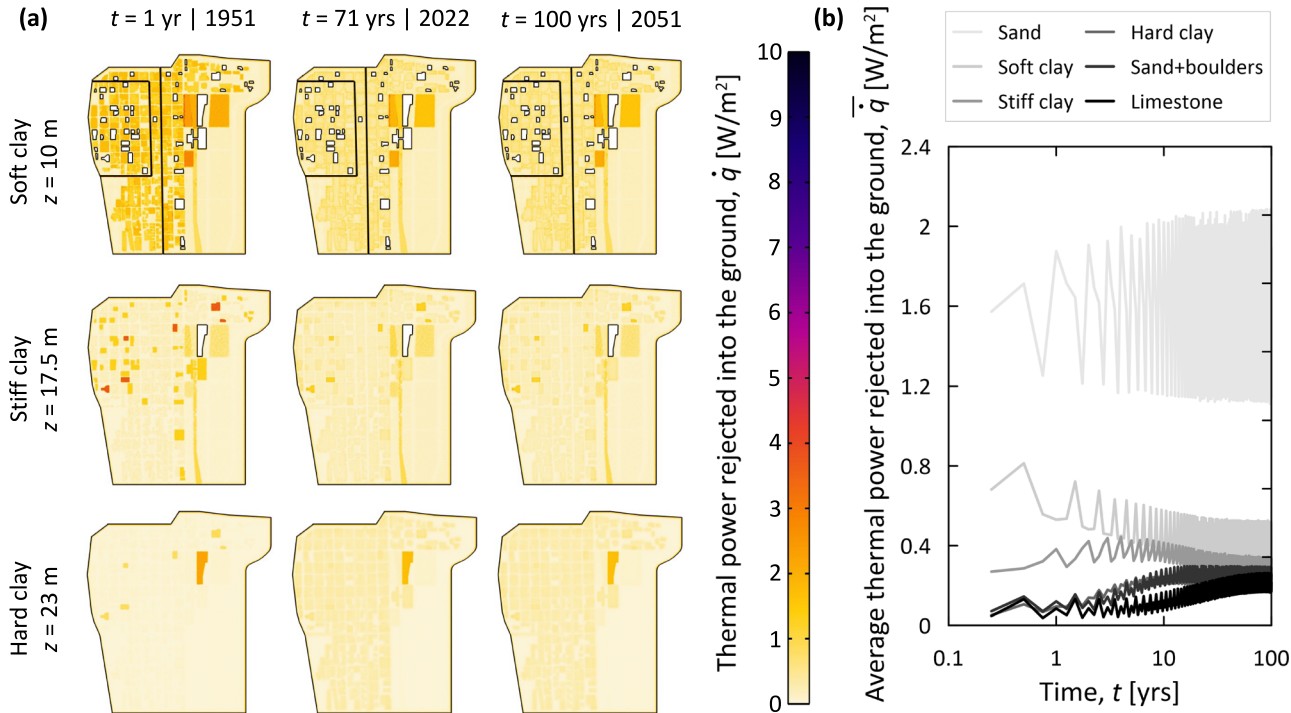

**Fig. 3 Waste heat per unit area of heat source injected in the subsurface of the Chicago Loop. a** Thermal power, $\dot{q}$, at varying depths, $z$, and times $t$ (the darker the shading, the higher the injected thermal power); **b** average trends of thermal power, $\bar{\dot{q}}$, for individual soil layers.

deformations and displacements caused by temperature variations may become excessive. These effects should be assessed on a case-by-case basis depending on the local temperature anomalies, the site ground properties, the geometrical features of the earth-contact structure or foundation system under consideration, the distribution of the applied loads, and the details of the reinforcement, among other variables. Existing built environments and historical buildings characterized by outdated or inappropriate construction features are not only those that contribute the most to the development of subsurface heat islands (due to customarily greater thermal losses in the ground than newer constructions) but are also those that can mostly suffer from ground deformations caused by such phenomena (due to a typically marked sensitivity to perturbations of their equilibrium, which is affected by temperature variations).

The results of this study indicate that the observed temperature rises in the subsurface of the Loop have caused negligible changes in the direction and magnitude of the groundwater flow. Furthermore, these results support that the Chicago River and Lake Michigan serve as buffers for the observed ground temperature rises, thus absorbing waste heat. The analysis of such an aspect remains outside the scope of this work but appears to deserve future consideration.

Temperature variations induced by subsurface heat islands are instantaneously limited but continuously rise and have not been considered in the design of existing underground structures and infrastructures, thus prone to unwanted thermally induced effects. Although the currently stable thermal state of the subsurface of the Chicago Loop district implies a small magnitude and limited future rise of thermally induced effects for the considered urban area, these effects may start or continue to affect other urban areas and cities unless subsurface heat islands are mitigated. Therefore, identifying and mitigating subsurface heat islands represent new priorities for urban planning strategies.

One approach to mitigate subsurface heat islands relies on the utilization of geothermal technologies to absorb waste heat on top of geothermal heat from the subsurface for use in buildings and

district energy networks. Another approach consists of retrofit interventions in underground environments (e.g., improvements of underground building envelopes and enclosures), which often lack thermal insulation and inject heat into the ground.

As can be noted in Fig. 3, noteworthy values of thermal power are continuously injected in the ground of the Loop: maximum values of thermal power per unit area of heat source can achieve almost 10 W/m$^2$ (Fig. 3a), whereas average values across the soil layers are typically lower than 1 W/m$^2$ (Fig. 3b). Harvesting such thermal power via geothermal technologies would contribute to only 0.5% of the annual heating consumption of the largest buildings of the Loop (i.e., 8 GWh vs. 1824 GWh as estimated through the simulations performed in this work and previously published data[56], respectively). Comparable results have been previously reported for other cities[43]. The limited size of heat-emissive surfaces in contact with the ground relative to the large size of the space conditioning surfaces of buildings in the Loop is the reason for the relatively small heating potential associated with the harvesting of waste heat in the considered district. However, this potential does not account for the inherent geothermal potential of the Loop's subsurface, which would indeed be harnessed through geothermal technologies and increase the total amount of heating energy that could be supplied to buildings. Therefore, harvesting waste heat from the ground will generally enhance the heating capacity of geothermal technologies, representing a valuable strategy to mitigate underground climate change in Chicago and other cities. If the deployment of geothermal technologies were to be impractical, the retrofit and thermal insulation of underground building enclosures would be a valid alternative strategy to mitigate underground climate change. Results such as those presented in this work can guide actions in both of these scenarios. The discussed results lead to the conclusion that subsurface heat islands represent not only a threat but also a resource for the decarbonization and sustainability enhancement of urban areas.

## Conclusions

This paper reveals a silent yet potentially problematic impact of subsurface urban heat islands on the performance of civil structures and infrastructures (e.g., building foundations, earth-retaining structures, and other underground structures and facilities). The root of this issue lies in thermally induced ground deformations and displacements, which develop slowly but continuously in the urban underground.

The ground deformations and displacements caused by subsurface heat islands can become an issue for the operational performance of structures and infrastructures with time, thus affecting their normal use and functionality. In other words, the development of such ground deformations and displacements does not threaten to lead to the collapse or rupture of structures and infrastructures but can potentially affect their durability, esthetic, and operational requirements. Therefore, the impacts of underground climate change on civil infrastructure do not threaten the safety of people, but they can potentially, and on a case-by-case basis, affect the efficient use and durability of such constructions, and consequently the comfort of people thriving therein.

The spatial and temporal evolution of subsurface heat islands is characterized by an inherent complexity, which depends on the evolutionary features of cities (urban morphology, urban infrastructure use, development, etc.). Despite such complexity, the results of this work support that the impacts of subsurface heat islands can be predicted realistically, promising to inform future urban planning strategies that can hamper these pervasive phenomena with effective and relatively simple strategies discussed in this study. From this perspective, subsurface heat islands can be considered a resource because they provide the opportunity to harness large quantities of waste heat that would otherwise be dispersed in the ground or, alternatively and in the first place, to minimize the loss of such heat via adequate retrofit interventions in buildings and infrastructures.

Future investigations at the intersection of urban science, mechanics, and energy are deemed necessary to expand the results provided by this exploratory work, in ways that can advance science, engineering, and technology and comprehensively inform revisions of urban planning strategies for different cities worldwide.

## Methods

**Temperature sensing network**. The temperature sensing network at the basis of this work, together with the Chicago Loop district where it is deployed, have been extensively characterized in other studies[22,56]. Therefore, their features are only described succinctly in this section.

Since 2019, temperature sensors have been deployed in a myriad of surface and subsurface environments in the Chicago Loop district, which is also known as "the cake" for its various levels of built environments above and below the ground surface. The aim of this endeavor has been threefold: (1) gather surface air temperature data for the downtown area of Chicago that can enrich complementary sensing endeavors for Chicagoland and Illinois[57,58], (2) establish relationships between the surface air temperature and the air temperature in underground built environments in the downtown area of Chicago[22], and (3) measure ground temperatures in the heart of Chicago[22].

Monitored surface environments consist of green parks and streets. Monitored subsurface environments include underground streets, building basements (serving residential, commercial, and tertiary buildings), private and public parking garages (including Millennium Garages—the largest underground parking system in North America, which comprises the four parking garages called Grant Park North, Millennium Park Garage, Millennium Lakeside, and Grant Park South), subway tunnels (the blue and red lines of the Chicago Transit Authority, CTA), a subsurface railway station (operated by the Chicago Metropolitan Rail system, METRA), a vast network of freight tunnels (non-operational since the 1992 Chicago flood), and the ground.

The Loop district presents a complex surface urban morphology. Taller and denser buildings characterize the northern portion of the Loop, although buildings become more widely spaced towards its north-eastern portion. In contrast, shorter and more widely spaced buildings characterize the southern portion of the Loop. Most buildings in the considered urban area serve commercial activities, whereas only a few (about one-third) serve residential uses. Four main green areas characterize the Loop: Millennium Park and Grant Park (the largest parks in the considered district), which are located on its east side, run along most of Michigan Avenue in this area and host Millennium Garages; Lakeshore East Park, which is located on the north-eastern portion of the district; and Dearborn and Roosevelt Parks, which are located on the south-western portion of the district.

Figure 4 presents a summary of air temperature data collected to date for surface and subsurface environments with the sensing network. Sensor locations are also provided for reference. Figure 4a illustrates monthly average surface air temperature data collected across the Loop with the deployed sensing network from 2020 till the end of 2022. These data are compared with monthly maximum, minimum, and average surface air temperature data collected at Chicago O'Hare airport from 1951 till the end of 2022[57]; monthly average water temperature data collected for Chicago River from 1951 till the end of 2022[57]; and monthly average grass and street temperature data derived with analytical expressions[59,60] from the monitored surface temperatures. Figure 4b illustrates the relationships between surface and subsurface air temperatures for building basements, Millennium Garages, and CTA tunnels established through the sensing network from 2019 till the end of 2022.

As can be noted in Fig. 4a, the monthly average surface air temperature markedly fluctuates over the year due to the harsh climate of Chicago. The recently monitored temperatures for the Loop are relatively close to the temperatures measured at Chicago O'Hare airport over the past 70 years. The monthly average temperatures determined for the water, grass, and streets in the Loop are temporally shifted and damped in magnitude compared to the average surface air temperatures.

As can be noted in Fig. 4b, the relationships between the surface and subsurface air temperatures for the monitored built environments can be described by linear functions. These functions allow for the prediction of air temperatures underground for given surface temperatures.

The gathered temperature data represent a unique resource for the study of subsurface heat islands. The reason is that they allow simulating with appropriate digital tools the waste heat emissions diffusing in the subsurface from the ground surface and underground built environments, with the promise to assess their impacts on the performance of civil infrastructure. This endeavor is performed in this study with a digital tool described in the following section.

**Numerical model and simulation**. The 3-D computer model of the Chicago Loop district underlying this study represents a digital twin of such an urban area. This facility has been built through an extensive characterization of the Loop from architectural, structural, hydrogeological, energy, and urban perspectives through site explorations and surveys, interactions with local companies, and the literature.

Figure 5 illustrates the 3-D model of the Loop. Such a model reproduces the myriad of underground built environments and the vast network of green spaces and streets that currently characterize the considered urban area. Based on data made available by the Illinois State Geological Survey[61], the model considers a horizontal soil stratigraphy composed of layers of sand, soft clay, stiff clay, hard clay, sand and boulders, and dolomitic limestone bedrock. The sand layer extends from the ground surface down to a depth of $z = 4$ m and has hence a thickness of $z_t = 4$ m. The soft clay extends between depths of $4 \leq z < 16$ m and has a thickness of $z_t = 12$ m. The stiff clay extends between depths of $16 \leq z < 19$ m and has a thickness of $z_t = 3$ m. The hard clay extends between depths of $19 \leq z < 27$ m and has a thickness of $z_t = 8$ m. Sand and boulders are found between depths of $27 \leq z < 34$ m, thus having a thickness of $z_t = 7$ m. Dolomitic limestone is found under such a layer. Groundwater is found at a depth of $z = 4$ m.

An analysis of historical documents and field surveys suggests that building basements penetrate the ground down to four characteristic depths: $z = 4$, 8.3, 12.6, and 17.2 m. On average they run at a depth of $\bar{z} = 6.2$ m. The depths of the parking environments that constitute Millennium Garages have been defined via the analysis of the architectural drawings of such environments, running at an average depth of $\bar{z} = 10$ m. The red and blue lines of the CTA subway, and the freight tunnels that run underneath almost every street in the Loop, are located at depths of $\bar{z} = 11$ and 12 m based on historic data[55], respectively. The subway tunnels are characterized by a cylindrical cross-section with a diameter of 4 m. The freight tunnels are egg-shaped and characterized by a height of 2.3 m and a width of 1.8 m[55]. Although the geometry of foundation systems underneath various buildings in the Loop has been characterized through the help of local engineering firms, the model does not reproduce such information because of its massive size. The model also neglects the presence of sewer and piping systems that are renowned for running at shallow depths in the Loop because exact information about their location was not made available by local entities for liability reasons.

The developed numerical model is used to run 3-D, time-dependent, thermo-hydro-mechanical finite element simulations. As a result, not only this model allows simulation of the heat that diffuses in the ground from the surface of the Loop and its various heat-emissive built environments; this model also allows simulation of the impacts of waste heat emissions on the deformation and the groundwater regime of the subsurface. Simulations are run with COMSOL Multiphysics (v. 5.5)[62].

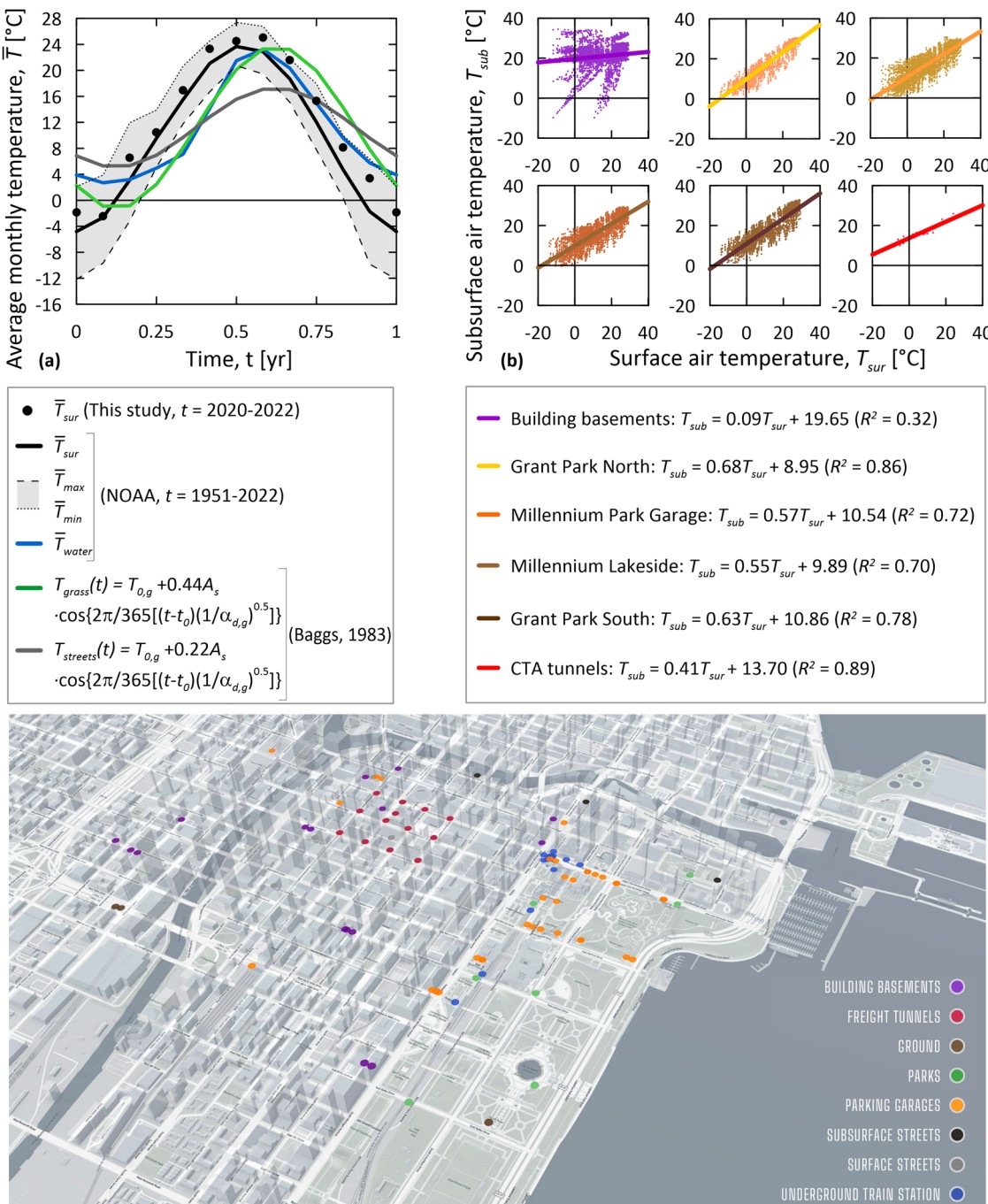

**Fig. 4 Temperature data monitored across the Chicago Loop. a** Average monthly temperature trend, $\bar{T}$, over time, $t$ ($\bar{T}_{sur}$: monthly average surface air temperature monitored by the developed sensing network and NOAA[57]; $\bar{T}_{max}$ and $\bar{T}_{min}$: maximum and minimum values of monthly average surface air temperature monitored by NOAA[57]; $\bar{T}_{water}$: monthly average water temperature monitored by NOAA[57]; $T_{grass}$ and $T_{streets}$: grass and street temperature values determined through the analytical expression proposed by Baggs[59] and the coefficients reported by Jense-Page et al.[60], where $T_{0,g}$ is the initial undisturbed ground temperature, $t_0$ is the time in days until the time the minimum air temperature from January 1st is measured for any considered location, $A_s$ is the amplitude of the annual temperature variation measured for any considered location, and $\alpha_{d,g}$ is the ground thermal diffusivity); **b** Relationship between surface air temperature, $T_{sur}$, and subsurface air temperature, $T_{sub}$, for different underground built environments across the Loop ($R^2$: coefficient of determination). The 3-D view of the sensing network has been created with a baseline image provided by OpenStreetMap.

The mathematical formulation employed for the simulation has already been presented in other studies[63,64] and is capable of reproducing complex problems of heat transfer, mass transfer, and deformation similar to the one addressed in this work. Therefore, the details of such formulation are not reported here for concision. From a qualitative perspective, the formulation resolves a conductive–convective energy conservation equation to reproduce the heat transfer in the ground, which is coupled with the momentum equilibrium equation to address the mechanics of the problem and the mass conservation equation to

simulate the presence and influence of groundwater flow. The idealization and assumptions governing the developed simulations are as follows[48]:

(i) Conduction heat transfer characterizes the ground in the shallowest 4 m of dry sand. Convection heat and mass transfers characterize the ground beyond 4 m of depth.

(ii) The ground is assumed to be isotropic, homogeneous, and characterized by a linear thermo-elastic behavior. The displacements and deformations of the

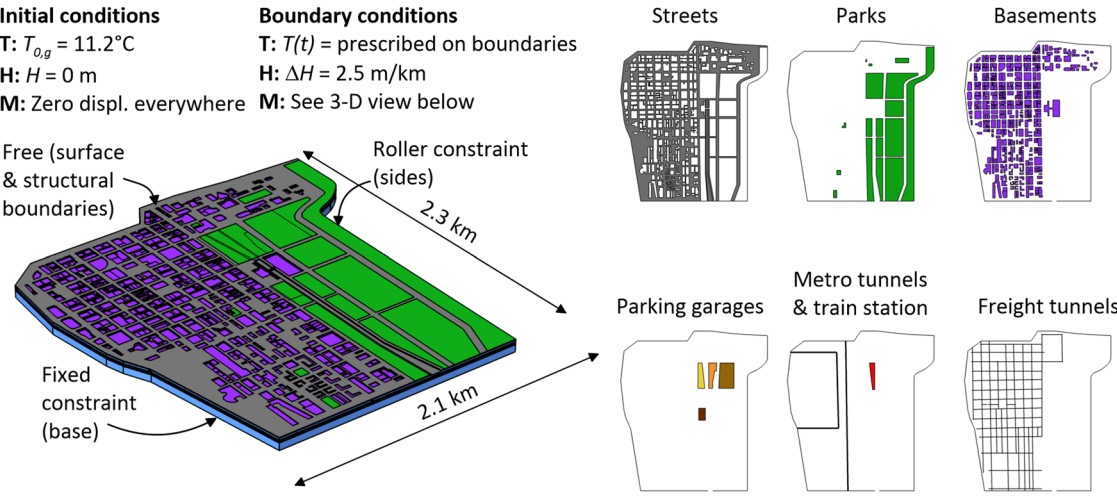

**Fig. 5 3-D computer model of the Chicago Loop.** The different colors represent different environments characterized by specific initial and boundary conditions ($T_{0,g}$: initial ground temperature; $H$: hydraulic head; $t$: time; $\Delta H$: change in head per unit reference length).

ground are described via a linear kinematic approach under quasi-static conditions. The materials that constitute the ground are characterized by pores filled with a fluid (e.g., water or air) and have thermo-physical properties given by the fluid and the solid phases.

(iii) The master equations governing the heat transfer, mass transfer, and deformation phenomena (i.e., continuity equation, momentum equation, and energy conservation equation, respectively) are coupled numerically in a time-dependent framework.

The employed modeling approach to simulate the heat transfer, mass transfer, and deformations may be considered advanced. However, it still incorporates simplifications discussed hereafter.

Probably the most significant simplification in the simulation of the heat and mass transfers that characterize the subsurface of the Loop lies in the fact that the model refers to the current urban morphology, although simulations are run from the 1950s till the 2050s. The heat and mass transfers characterizing the subsurface of urban areas are influenced by the evolutionary features of such complex environments (e.g., the construction and demolition of built environments, the transient loads acting on such environments, the variable and heterogeneous environmental conditions at the surface, etc.), and depend on the actual, spatially non-uniform properties of the ground. However, such aspects are daunting to reproduce with digital tools, especially when considering extensive urban areas such as the Loop. Fortunately, an analysis of historical data shows that the number and location of buildings across the Loop have not significantly changed since the 1950s. New buildings have indeed been constructed in the meantime but in most cases over the footprint of previous buildings. Therefore, it is argued that the urban morphology considered in the model, while static, provides an adequate representation of buildings and underground built environments across the Loop. The close comparison between the temperature data obtained for the ground through the developed numerical simulations and the sensing network supports that the model can well reproduce the heat and mass transfer characterizing the Chicago Loop district with the considered assumptions (see "Methods, Numerical model validation").

Probably the most significant simplification in the simulation of the deformation of the Loop lies in the employed thermo-elastic assumption. When soils are subjected to thermal, hydraulic, and mechanical loads, they can be characterized by irreversible (i.e., plastic) deformations, which are typically non-linear. The actual stress state and history of soils also crucially govern their behavior. However, strains in the ground increase as stresses increase, and linear elastic theory has been proven sufficiently accurate for scientific and engineering purposes, provided that appropriate material parameters are employed[50]. In this context, the availability of a substantial amount of field and laboratory test results[53,65–71] for the soil layers beneath the Loop provides confidence in the available material parameters, thus supporting the use of a linear thermo-elastic approach that has been proven capable of capturing thermally induced deformations of civil structures and infrastructures in other studies[48]. By definition, this modeling approach does not account for viscous (e.g., creep) effects, which are renowned to characterize soils and construction materials and involve time-dependent surges in deformations and displacements under constant applied loads. Meanwhile, this approach appears valuable because of two reasons. First, it provides ground deformations and displacements that implicitly account for the stiffening effect of foundations, which have been neglected in the simulation because of the large size of the studied problem. Second, it provides ground deformations and displacements that appear reliable in magnitude, without representing an unjustified source of concern. The detailed resolution of the

considered problem, together with the complexity of the analyses performed, further justifies using a linear thermo-elastic approach, which would incorporate undue complexity otherwise.

The material parameters used in the simulations are summarized in Tables 1–3 with corresponding bibliographic sources. The initial and boundary conditions considered in the simulations are summarized in Fig. 5.

Thermal initial conditions consist of a uniform ground temperature of $T_{0,g} = 11.2\,°C$ as per the data gathered through the deployed sensing network[22]. Mechanical initial conditions consist of zero initial displacements or applied forces. Hydraulic initial conditions consist of a constant hydraulic head $H = 0$ m.

Thermal boundary conditions consist of the following: a time-varying temperature boundary condition imposed on the uppermost surface of the model for the streets and green spaces (following the model of Baggs[59] and the surface air temperature data provided by the National Oceanic and Atmospheric Administration, NOAA, from 1951 to 2051[57]—see Fig. 4a); a time-varying temperature boundary condition for the vertical surfaces of the model (following the Chicago River water data provided by NOAA from 1951 to 2051[57]—see Fig. 4a); a fixed constant temperature for the bottom surface of the model $T_{0,g} = 11.2\,°C$; and a time-varying temperature for the interfaces between the underground built environments and the ground (following the relationships between surface and subsurface air temperatures collected through the sensing network—see Fig. 4b). These fixed temperature boundary conditions (i.e., Dirichlet boundary conditions) involve a heat transfer into/from the enclosed material volume(s) according to the imposed temperature on the surface(s) of the volume(s) and the temperature of the volume(s). These boundary conditions do not consider (by definition) any convection or radiation effects, which may characterize some underground built environments (e.g., due to airflows caused by the movement of trains in tunnels) and the ground surface (e.g., due to solar radiation). Although potentially approximate in instances, this approach is motivated by the lack of data about these phenomena over the analyzed timeframe (from the 1950s till the 2050s). Nevertheless, this approach appears acceptable for the analysis of ground temperature anomalies in the considered urban area due to the close comparison between the modeled and measured temperature data, both at shallow and relatively deep locations (see "Methods, Numerical model validation"). At shallow depths, convection and radiation effects might be present but appear negligible compared to other aspects of the problem that have been considered in the simulations. At depth, convection and radiation effects arguably characterize a minimal proportion of the underground built environments in the Loop, which mostly consist of building basements (typically not affected by airflows or extreme heat sources) and only include two train tunnels (where airflows are present). Overall, these effects thus appear negligible for the subsurface of the Loop and insignificant at the monitored locations.

Mechanical boundary conditions consist of free displacements on the top surface of the model and any surface between underground built environments and the ground. In contrast, displacements on the external vertical and bottom surfaces of the model are fixed according to the roller and pinned conditions, respectively.

Hydraulic boundary conditions consist of an imposed hydraulic gradient of $\Delta H = 2.5$ m/km as per field explorations[72]. The considered hydraulic gradient is thus minimal. For the sake of general information, simulations performed without due account of such phenomenon have yielded markedly close results to those presented in this study.

Simulations are performed over a timeframe of $t = 100$ years. Data are saved every month during this timeframe. All values of compressive stresses, contractive

**Table 1 Thermo-physical properties of the ground underneath the Chicago Loop district.**

| Property class | Geological layer | Thermal Conductivity, $\lambda$ [W/(m °C)] | Source | Heat capacity at constant pressure, $c_p$ [J/(kg °C)] | Source | Density, $\rho$ [kg/m³] | Source |
|---|---|---|---|---|---|---|---|
| Thermal/ Physical | Sand | 1 | Assumed from Laloui and Rotta Loria[48] | 782 | Assumed from Laloui and Rotta Loria[48] | 1918 | Given by Finno et al.[73] |
| | Soft clay | 1.22 | Calculated from Saxena and Militsopoulos[66] | 1456 | Given by Saxena and Militsopoulos[66] | 1846 | |
| | Stiff clay | 1.22 | | 1456 | | 2000 | |
| | Hard clay | 1.22 | | 1456 | | 2081 | |
| | Sand and boulders | 3 | Assumed from Laloui and Rotta Loria[48] | 1035 | Assumed from Laloui and Rotta Loria[48] | 2320 | Given by Manger[74] |
| | Dolomitic limestone (bedrock) | 3 | | 835 | | 2639 | |

**Table 2 Thermo-mechanical properties of the ground underneath the Chicago Loop district.**

| Property class | Geological layer | Young's modulus, $E$ [MPa] | Source | Poisson's ratio, $\nu$ [−] | Source | Linear thermal expansion coefficient, $\alpha$ [°C⁻¹] | Source |
|---|---|---|---|---|---|---|---|
| Mechanical | Sand | 31 | Given by Finno and Calvello[75] | 0.2 | Given by Finno and Calvello[75] | $1.0 \times 10^{-5}$ | Assumed from Laloui and Rotta Loria[48] |
| | Soft clay | 10 | Given by Finno and Calvello[75] | | Assumed from Laloui and Rotta Loria[48] | $-9.0 \times 10^{-6}$ | Determined from Saxena and Militsopoulos[66,a] |
| | Stiff clay | 31 | | 0.2 | | $-9.0 \times 10^{-6}$ | |
| | Hard clay | 214 | | | | $9.0 \times 10^{-6}$ | Given by Saxena and Militsopoulos[66] |
| | Sand and boulders | 75 | | | | $1.0 \times 10^{-5}$ | Assumed from Laloui and Rotta Loria[48] |
| | Dolomitic limestone (bedrock) | 45,586 | Calculated from Hamwey and Naus[76] | 0.1 | Calculated from Shalabi et al.[77] | $2.2 \times 10^{-6}$ | Calculated from Lamar[65] |

ᵃThe negative values of the thermal expansion coefficient involve a volumetric contraction of the corresponding material upon heating. Such values are considered for the normally consolidated clayey layers because they suffer from the thermal collapse phenomenon in light of their consolidation state[45]. Using a negative thermal expansion coefficient allows to representatively account for the positive bulk thermal contraction of soils due to the thermal collapse phenomenon[78].

**Table 3 Hydrogeological properties of the ground underneath the Chicago Loop district.**

| Property class | Geological layer | Hydraulic conductivity, $k$ [m/s] | Source | Porosity, $n$ [−] | Source | Overconsolidation ratio, $OCR$ [−] | Source |
|---|---|---|---|---|---|---|---|
| Hydraulic/ Geological | Sand | $1.74 \times 10^{-6}$ | Given by Finno and Calvello[75] | 0.35 | Given by Terzaghi et al.[79] | N.A. | |
| | Soft clay | $1.04 \times 10^{-9}$ | | 0.41 | Calculated from Finno et al.[73] | 1 | Given by Finno et al.[73] |
| | Stiff clay | $1.04 \times 10^{-9}$ | | 0.31 | | 1 | |
| | Hard clay | $1.04 \times 10^{-9}$ | | 0.15 | Calculated from Hamwey and Naus[76] | >3 | |
| | Sand and boulders | $1.04 \times 10^{-5}$ | Assumed from Laloui and Rotta Loria[48] | 0.21 | Given by Manger[74] | N.A. | |
| | Dolomitic limestone (bedrock) | $1.04 \times 10^{-12}$ | | 0.10 | | NA. | |

strains, downward displacements, and injected thermal powers are considered positive in this paper.

**Data analysis.** Data are exported with COMSOL Multiphysics (v. 5.5) and analyzed with Microsoft Excel (v. 16.68). Data are plotted with Grapher (v. 2022).

**Numerical model validation.** Figure 6 summarizes the validation of the modeling approach used in this study by comparing representative experimental data obtained from the field with numerical results. Figure 6a compares ground

temperature data obtained through the developed simulation for Grant Park in the Loop at depths $z = 0.1$, 0.2, and 4 m, with field data collected at shallow depths in St. Charles at depths $z = 0.1$ and 0.2 m and field data gathered through the sensing network in Grant Park at a depth $z = 4$ m. Figure 6b illustrates the temperature trend characterizing several locations in the heart of the Loop starting from 1951 till 2051 and compares them with temperature data gathered through the sensing network from 2020 till 2022 at corresponding locations.

As can be remarked from Fig. 6a, the temperature data predicted numerically and monitored experimentally for Grant Park at a depth of $z = 4$ m match well. The numerical and experimental temperature data referring to depths of $z = 0.1$

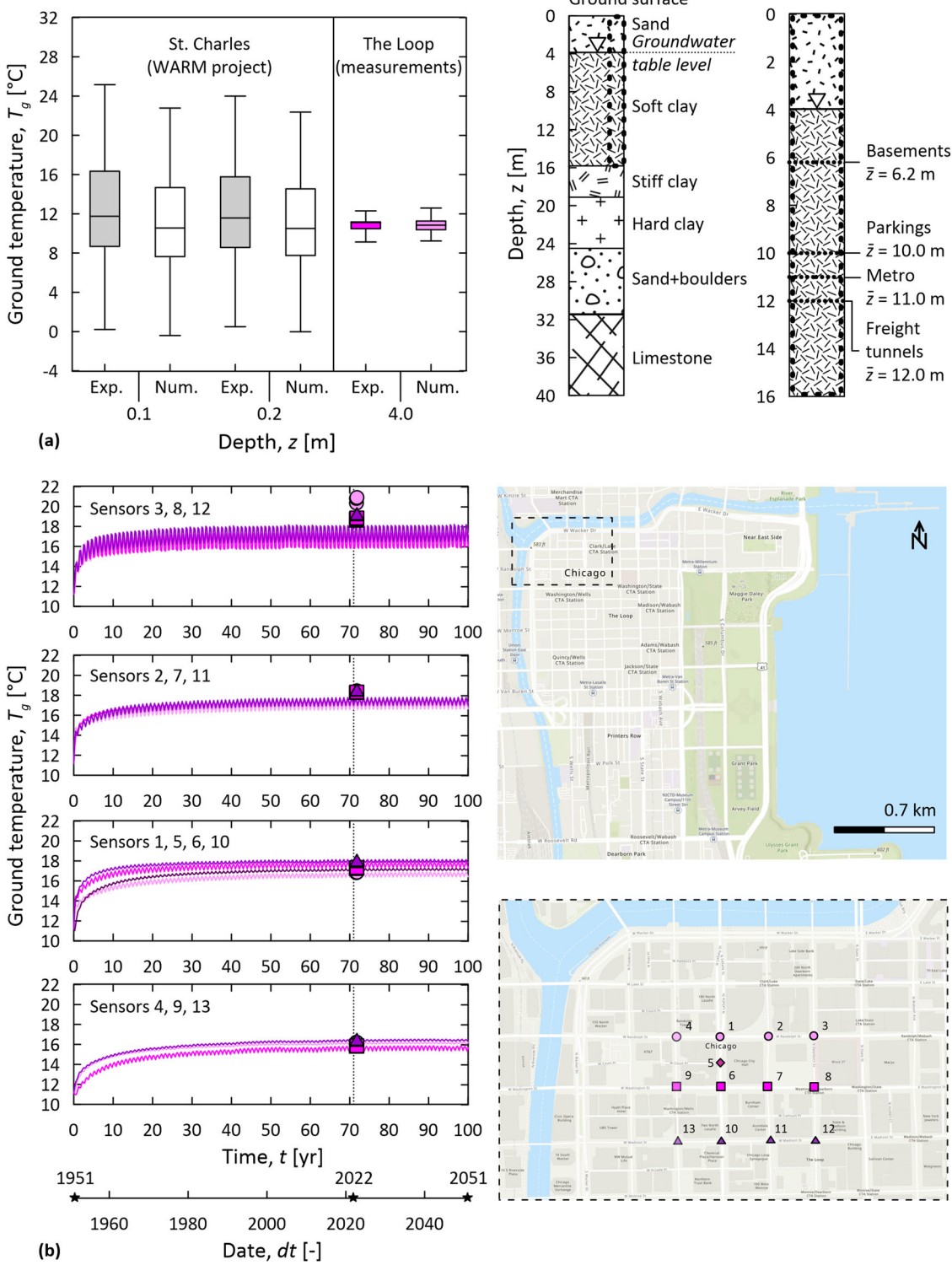

**Fig. 6 Validation of the numerical simulation results against experimental monitoring data. a** Comparison between the ground temperatures $T_g$ provided by the model in Grant Park and recorded at corresponding locations in St. Charles[58] and the same Park[22] (in the box plot: the center line indicates the median; the box edges indicate a 95% confidence level; the whiskers indicate maximum/minimum); **b** comparison between the ground temperature trends over time $t$ provided by the model in the heart of the Loop and the sensing network. The plan views illustrating the locations of the sensors have been created with baseline images provided by OpenStreetMap.

and 0.2 m are also markedly close, even though two different locations (i.e., St. Charles and the Loop) are considered. Such a result indicates comparable thermophysical properties for the ground at the considered locations, which may derive from the relatively uniform geology of Illinois because of the glacial formation of Michigan Lake.

As can be remarked from Fig. 6b, the temperature variations predicted numerically from 1951 to date for the different ground locations beneath the Loop at a depth of $z = 12$ m closely match the temperature data gathered in the field through the sensing network. Such a close comparison between the numerical and experimental data, similar to that reported in Fig. 6a, holds even when slightly different depths or different

thermo-physical properties (by 50%) may be considered for the ground in the numerical model, supporting the robustness of the considered simulation and its input data (e.g., material properties and boundary conditions).

Based on a sound analysis of the results of this work, it appears inappropriate to deterministically and unequivocally link the temperature variations provided by the developed simulation with exact dates in the past, present, or future due to the complexity of the considered problem. Nonetheless, it appears appropriate to argue that the developed simulation provides results representative of reality (with an accuracy of a few years for any studied date). Therefore, not only the developed numerical simulation appears capable of retrieving the temperature variations and their impacts on the subsurface of the Loop from the 1950s to current times but also of predicting the likely influence of underground climate change in such a district over the next 30 years. This highly satisfactory result strongly corroborates the representativeness of the simulations performed in this study.

## Data availability

## Code availability
The simulation at the basis of this work has been performed with the proprietary software COMSOL Multiphysics (v. 5.5)[62]. No custom code has been generated for the analysis of the simulation results. Information about the 3-D model of the Loop is available upon request from A.F.R.L.

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

## Acknowledgements

The author of this work would like to express his sincere gratitude to the following individuals and institutions for their willingness to facilitate this research through the provision of access to environments where the temperature sensors constituting the network employed in this study were installed: Patrick D. Martin from InterPark; Admir Sefo from Next Parking; Timothy G. Pitzen and Lynn M. Dyon from METRA Chicago; Ron Tabaczynski from the Building Owners and Managers Association of Chicago; J.J. Madia, Edwin Flores, and Matt Lewis from the Chicago Department of Transportation; Jim Rylowicz and Geoff Bares from Centrio Energy; Jamie Ponce and Firas Suqi from City Tech Collaborative; Brett Gitskin from ECS Midwest; Luda Chervona and Nawar Telche from Hotel Julian; Pierre Giacotto and Jeff Green from The Blackstone Hotel; Gary Platt and Brian Poirier from La Quinta Inn & Suites by Wyndham; Kelsey Brown and Guerlin Frederic from Lake & Wells Apartments; Kevin Hanley from Union League Club of Chicago; and Bruce Moffat. The help provided by Jennifer L. Kunde, Maggie Waldron, and Richard Cummo from Northwestern University to facilitate the deployment of the sensing network at the basis of this work is greatly appreciated. Anjali Thota and Xiaohui Gong are thanked for providing the author with updated temperature data at the time this manuscript was written. Stephan Harmann and Lisa Cassina are thanked for their contributions to the early developments of the computer model at the basis of this work. Lyesse Laloui is thanked for the stimulating discussions about this subject. The financial support provided by the Murphy Society and the Alumnae of Northwestern University to develop the sensing network used for this work is thankfully acknowledged. This work is further supported by the National Science Foundation under Grant No. 2046586.

## Author contributions

A.F.R.L. designed the research, performed the simulations and data analysis, created the figures, and wrote and revised the paper.

## Competing interests

The author declares no competing interests.
