## [Peer Review File · Communications Engineering]

Reviewers' comments:

Reviewer #1 (Remarks to the Author):

This is an excellent study that is certainly worthy of publication. However, the reviewer found the structure (of presenting results first and then the methodology and validation later) disconcerting. The authors need to preempt questions raised when reading the results by referring forward to later sections, so the reader is aware that questions they have in their minds will be answered later.

Specific comments

L28 trains braking (not breaking)

L64-65 most densely populated district

L112 "variable climate change" should be "variable temperature change"

Fig. 1a Please explain more of the geography of the "Chicago Loop" and the distributions of buildings and infrastructure represented in the figure. Also, say more about the geological profile of soils over bedrock (and their depths) (this could be by reference to later section)

The modelling must make assumptions on the amount of heat energy injected into the ground. On what basis were these assumptions made? Building by building or through some formula? More information of temperature boundary conditions would be helpful.

Weighted average temperature changes are close to those of the Limestone bedrock. What depth was used for averaging and how was weighting applied? In the 10-50 year period the soils seem to be warming more than the bedrock, but in the 71-100 year period, the reverse is true. This needs commenting on and explaining.

L91-119 The introduction suggests that numerical modelling is validated against actual measurements. Refer to the later section that provides the validation of the modelled results.

L141-150 Movements of 10s of mm could be problematic, but if relatively uniform are not likely to be so. It is differential movements that are likely to be problematic (Burland and Wroth, 1975). Some comment on differential movements would therefore be helpful.

L339-342 "close comparison between the temperature data obtained for the ground through the developed numerical simulations and the sensing network" - refer to where this validation is presented.

Reviewer #2 (Remarks to the Author):

This study quantifies the issues caused by subsurface heat islands and discusses their impacts on civil

infrastructure. The study performs 3-D computer simulations, which are informed and validated by data collected via a wireless sensing network. The work adopts the Chicago Loop district as a case study. The obtained results highlight that the underground climate change in the Chicago Loop represents a silent hazard for civil infrastructures, which is likely to be applicable to other urban areas worldwide. The study is relevant and timely, as well as novel. The paper is well-structured and written. Figures and tables are appropriate. My only concern regards the case study, which is limited to the Chicago Loop district, and does not include a wider area and/or multiple urban areas. If the editors think that this is suitable for Nature Comms Eng, I am supportive for this article to be published after addressing minor comments (below).

L46: it would be good to provide stronger motivation, e.g. what are these “issues” affecting infrastructure (in addition to the two brief examples)? Could more examples be provided, with perhaps evidence from literature/recent events?

L47: the use of “meanwhile” throughout the paper is not always correct. “Meanwhile” means “in the meantime” (and the sentence should make sense if substituted with it). In this case, replace “Meanwhile” with “On the contrary”. Please, check all the other cases (they are many).

L59-60: improve readiness. “...design; however, no existing...” (replace “meanwhile”).

L139 and other lines: “thermally-induced”.

L152-154: already said.

L155-160: it would fit best in the Introduction.

L161: repetition.

L190: improve readiness, “The results led to the conclusion that...”.

L213: substitute “elsewhere” with “in the literature”.

L214: replace “the following” with “this section”.

L216: delete “densest...Chicago”, said already.

L376: replace “Everywhere ...paper” with “All values of compressive...” and add “in this paper” at the end of the sentence.

L423-424: I cannot follow this sentence which reads as “this result corroborates the results”. Please, rephrase.

The silent impact of underground climate change on civil infrastructure

Alessandro F. Rotta Loria^{1*}

¹Mechanics and Energy Laboratory, Department of Civil and Environmental Engineering, Northwestern University, 2145 Sheridan Road, Evanston, IL 60208, USA

*Corresponding author: af-rottaloria@northwestern.edu

Response to Reviewers' comments

Reviewer #1

General comment: This is an excellent study that is certainly worthy of publication. However, the reviewer found the structure (of presenting results first and then the methodology and validation later) disconcerting. The authors need to preempt questions raised when reading the results by referring forward to later sections, so the reader is aware that questions they have in their minds will be answered later.

General response: Thank you for your positive comments about this work. The author agrees with the fact that the structure of the present paper (with results presented before the methodology section) differs from other journals. However, this structure is required by *Communications Engineering*. As suggested by the Reviewer, additional references to specific sections of the paper that include key considerations for the understanding of the work have been reported as appropriate to maximize the clarity of this study since the start of the manuscript.

C1: L28 trains braking (not breaking)

R1: Apologies for the oversight, and thank you. The mistake has been corrected (p. 2).

C2: L64-65 most densely populated district

R2: Thank you for the suggestion. The text has been amended (p. 3).

C3: L112 “variable climate change” should be “variable temperature change”

R3: Thank you for the suggestion. The text has been amended (p. 6).

C4: Fig. 1a Please explain more of the geography of the “Chicago Loop” and the distributions of buildings and infrastructure represented in the figure. Also, say more about the geological profile of soils over bedrock (and their depths) (this could be by reference to later section)

R4: Thank you for the suggestions. Details about the morphology of the Loop have been included in “Methods, Temperature sensing network” (p. 14), and the description of the predicted ground temperature variations has been expanded in “Results” (p. 5 and 6). Further comments about the geological profile have also been reported in “Results” (p. 6), and reference to the section “Methods, Numerical model and simulation,” where the details of the soil stratigraphy are presented, has been made for clarity.

C5: The modelling must make assumptions on the amount of heat energy injected into the ground. On what basis were these assumptions made? Building by building or through some formula? More information of temperature boundary conditions would be helpful.

R5: In the analyses, no assumption has been made about the amount of heat injected into the ground. This aspect is calculated in the simulations by resolving the heat transfer problem for the imposed boundary conditions. In alignment with the Reviewer’s comment, the description of the applied thermal (and hydraulic) boundary conditions has been expanded to clarify the rationale and appropriateness of the employed modeling approach (p. 21 and 22).

C6: Weighted average temperature changes are close to those of the Limestone bedrock. What depth was used for averaging and how was weighting applied? In the 10-50 year period the soils seem to be warming more than the bedrock, but in the 71-100 year period, the reverse is true. This needs commenting on and explaining.

R6: As now specified in the manuscript, temperature values in Fig. 1(a) (and displacement values in Fig. 2(b)) are averaged over the volume of the relevant soil layer. The weighted ground temperature is an average of these values and weighting is applied with respect to the thickness of the soil layers. It is worth noting that, by removing the last layer of limestone from the calculation of the weighted average, a value of 0.50°C/yr is obtained from years 1 to 50 (instead of 0.49°C/yr when this layer is considered), whereas a value of 0.17°C/yr is obtained for years 71 to 100 (instead of 0.14°C/yr); therefore, it is a coincidence that the weighted ground temperature for the entire soil volume is close to the average ground temperature for the limestone layer. As suggested by the Reviewer, the description of the heating rates has been expanded (p. 6-7).

C7: L91-119 The introduction suggests that numerical modelling is validated against actual measurements. Refer to the later section that provides the validation of the modelled results.

R7: Thank you for this suggestion. The change has been implemented (p. 4).

C8: L141-150 Movements of 10s of mm could be problematic, but if relatively uniform are not likely to be so. It is differential movements that are likely to be problematic (Burland and Wroth, 1975). Some comment on differential movements would therefore be helpful.

R8: Thank you for pointing this out – it fully agreed on such consideration. To reflect this aspect in the paper, the digression on potentially problematic ground displacements has been expanded (see the revised section “Underground climate change: a silent hazard which can represent a resource” – p. 9 and 10). Furthermore, a specific comment on the magnitude of differential displacements predicted by the model has been reported (see the end of the section “Deformations caused by the underground climate change in the Loop” – p. 8). For the information of the Reviewer, numerical simulations of the mechanical behavior of individual building foundations subjected to waste heat are being performed to investigate this problem with due account of the features of foundation systems and superstructures, thanks to data gathered from local construction companies.

C9: L339-342 “close comparison between the temperature data obtained for the ground through the developed numerical simulations and the sensing network” - refer to where this validation is presented.

R9: Thank you for this suggestion. The change has been implemented (p. 19).

Reviewer #2

General comment: This study quantifies the issues caused by subsurface heat islands and discusses their impacts on civil infrastructure. The study performs 3-D computer simulations, which are informed and validated by data collected via a wireless sensing network. The work adopts the Chicago Loop district as a case study. The obtained results highlight that the underground climate change in the Chicago Loop represents a silent hazard for civil infrastructures, which is likely to be applicable to other urban areas worldwide.

The study is relevant and timely, as well as novel. The paper is well-structured and written. Figures and tables are appropriate. My only concern regards the case study, which is limited to the Chicago Loop district, and does not include a wider area and/or multiple urban areas. If the editors think that this is suitable for Nature Comms Eng, I am supportive for this article to be published after addressing minor comments.

General response: Thank you very much for the positive assessment of this work. The Author rejoices to hear this feedback. It is agreed that the work focuses on one urban district only and information about other urban areas would be valuable. However, the reported results are still considered a substantial and informative contribution, which resort to more than four years of work during which the temperature sensing network and the 3-D finite element analyses that are at the basis of this study have been developed. At the time of writing, the author of this work is unable to perform a similar study on other urban areas, mainly because he does not have the necessary data to perform a comparable investigation for other urban areas. Nevertheless, by providing unprecedented results on the silent impact of subsurface heat islands on the performance of civil infrastructures, this study not only presents novel scientific results that can

inform the scientific community, the engineering practice, and other individuals and entities dealing with cities, but also foster and inform future studies that may lead to the creation of a new research field in the years to come. Therefore, it is hoped that this investigation will be considered appropriate for publication in *Communications Engineering*.

C1: L46: it would be good to provide stronger motivation, e.g. what are these “issues” affecting infrastructure (in addition to the two brief examples)? Could more examples be provided, with perhaps evidence from literature/recent events?

R1: Thank you for the suggestion. Additional comments on the issues that can characterize transportation infrastructure due to subsurface heat islands have been provided, together with corresponding references (p. 2 and 3).

C2: L47: the use of “meanwhile” throughout the paper is not always correct. “Meanwhile” means “in the meantime” (and the sentence should make sense if substituted with it). In this case, replace “Meanwhile” with “On the contrary”. Please, check all the other cases (they are many).

R2: Thank you very much for this correction. The text has been revised as suggested (p. 3). Furthermore, the use of “meanwhile” has been revised everywhere in the manuscript.

C3: L59-60: improve readiness. “...design; however, no existing...” (replace “meanwhile”).

R3: Thank you – amended as suggested (p. 3).

C4: L139 and other lines: “thermally-induced”.

R4: Thank you for this suggestion. However, consistent with previous publications, it is preferred to write “thermally induced.”

C5: L152-154: already said.

R5: Thank you for this comment. The considered paragraph has been deleted.

C6: L155-160: it would fit best in the Introduction.

R6: Thank you for this suggestion. However, it is preferred to keep this digression in the subsection under consideration without moving it into the Introduction.

C7: L161: repetition.

R7: Thank you for the recommendation. The considered paragraph has been revised (p. 10 and 11).

C8: L190: improve readiness, “The results led to the conclusion that...”.

R8: Thank you for the suggestion. The text has been amended (p. 11).

C9: L213: substitute “elsewhere” with “in the literature”.

R9: Thank you for the suggestion. The text has been amended (p. 13).

C10: L214: replace “the following” with “this section”.

R10: Thank you for the suggestion. The text has been amended as suggested (p.13).

C11: L216: delete “densest...Chicago”, said already.

R11: Thank you for the suggestion. The considered text has been removed (p.13).

C12: L376: replace “Everywhere ...paper” with “All values of compressive...” and add “in this paper” at the end of the sentence.

R12: Thank you for the suggestion. The text has been amended as suggested (p. 21).

C13: L423-424: I cannot follow this sentence which reads as “this result corroborates the results”. Please, rephrase.

R13: Thank you for the suggestion. The considered text has been amended (p. 26).

REVIEWERS' COMMENTS:

Reviewer #1 (Remarks to the Author):

The changes made by the author are satisfactory and I would be happy for this to go to publication.

One minor correction in the last line above Figure 1. It should be "bedrock is further away" rather than "bedrock is father away".

Reviewer #2 (Remarks to the Author):

All comments have been addressed

The silent impact of underground climate change on civil infrastructure

Alessandro F. Rotta Loria^{1*}

¹Mechanics and Energy Laboratory, Department of Civil and Environmental Engineering, Northwestern University, 2145 Sheridan Road, Evanston, IL 60208, USA

*Corresponding author: af-rottaloria@northwestern.edu

Response to Reviewers' comments

Reviewer #1

Comment: The changes made by the author are satisfactory and I would be happy for this to go to publication. One minor correction in the last line above Figure 1. It should be "bedrock is further away" rather than "bedrock is father away".

Response: Thank you for your positive comments about this work. The author has amended the text as suggested.

Reviewer #2

Comment: All comments have been addressed

Response: Thank you.